**Guglielmin Mauro[1]\*, Donatelli Marco[2], Semplice Matteo[3], Serra Capizzano Stefano[2,4].**

1\* Department of Theoretical and Applied Sciences, Insubria University, Via Dunant 3, 21100 Varese
mauro.guglielmin@uninsubria.it;

2 Department of Science and High Technology, Insubria University; 3 Department of Mathematics, University of Turin; 4 Department of Information Technology, Uppsala University.

# GROUND SURFACE TEMPERATURE RECONSTRUCTION FOR THE LAST 500 YEARS OBTAINED FROM PERMAFROST TEMPERATURES OBSERVED IN THE STELVIO SHARE BOREHOLE, ITALIAN ALPS.

## ABSTRACT

Here we present the results of the inversion of a multiannual temperature profile (2013, 2014, 2015) of the deepest borehole (235 m) in the mountain permafrost of the world located close to Stelvio Pass in the Central Italian Alps. The Stelvio Share Borehole (SSB) is monitored since 2010 with 13 thermistors placed at different depths between 20 and 235 m. The negligible porosity of the rock (dolostone, < 5%) allows to assume the latent heat effects also negligible. The inversion model here proposed is based on the Tikhonov regularization applied to a discretized heat equation, accompanied by a novel regularizing penalty operator. The general pattern of ground surface temperatures (GST) reconstructed from SSB for the last 500 years are similar to the mean annual air temperature (MAAT) reconstructions for the European Alps. The main difference with respect to MAAT reconstructions relates to post Little Ice Age (LIA) events. Between 1940 and 1989, SSB data indicate C a cooling of ca 1°C. Subsequently, a rapid and abrupt GST warming (more than 0.8°C per decade) was recorded between 1990 and 2011. This warming is of the same magnitude as the increase of MAAT between 1990 and 2000 recorded in central Europe and roughly doubling the increase of MAAT in the Alps.

## 1 INTRODUCTION

The thermal regime of the uppermost ground is determined by the geothermal heat flow and by the fluctuations of temperature at the surface. If rock was homogeneous and no temperature change were to occur at the surface, the temperature would increase linearly with depth, unless spontaneous heat production is present on the vicinity of the well.. The gradient of this temperature increase would be governed solely by the magnitude of the terrestrial heat flow and by the thermal conductivity of the rock. However, variations of ground surface temperature (GST) propagate downwards into the rock as attenuating thermal waves, superimposed on the aforementioned linear temperature profile. The depth to which disturbances can be recorded is determined mainly by the amplitude and duration of the temperature change at the surface. Generally, propagation of climate signals is slow and it can take more than 1,000 years to reach the depth of 500m (Huang et al., 2000). For a better conservation of the climate

signal in the thermal profile, no lateral heat advection (due for example to ground water flow) should be
present (Lewis and Wang, 1992). Since normally no groundwater circulation is present within continuous
permafrost in the polar areas but also in rocky areas within mountain permafrost, boreholes drilled in these
areas are particularly suited for GST reconstructions.
Lachenbruch and Marshall (1986) were among the first to demonstrate that thermal profiles obtained
from boreholes drilled in permafrost can be used to reconstruct ground surface temperature changes.
These do not require calibration because the heat conduction equation is directly used to infer
temperature changes at the ground surface. Today, the majority of permafrost boreholes used to
reconstruct ground surface temperatures are located in the Polar regions of North America and Eurasia
where the boreholes can be drilled on flat terrain, with negligible topographical effects, and with a
permafrost thicknesses typically exceed 100 m, thereby providing deep temperature logs and long ground
surface temperature reconstructions. On the other hand several factors like porosity, water/ice and latent
heat flows can influence significantly the thermal properties and the thermal signal especially measured in
frozen sediments boreholes as well discussed in Mottaghy and Rath (2006).
The Share Stelvio borehole (SSB) in the Italian Alps is the deepest drilled within permafrost in the mid-
latitude mountains of Europe. Because the permafrost thickness exceeds 200 m at this site it allows
reconstruction of GST for some centuries and much more than in the other mountain permafrost
boreholes.. In addition, the Stelvio borehole is located on a rounded summit with gentle side slopes.
Therefore, site-specific topographic influences are largely eliminated. As such, it is different to the other
boreholes drilled in permafrost in the Alps (e.g. PACE boreholes at Schilthorn or Stockhorn; see Harris et al.,
2003; Gruber et al.,2004; Hilbich et al., 2008; Harris et al., 2009).
Recent atmospheric warming (over the last century) in the European Alps has been roughly twice the global
average (Böhm et al., 2001; Auer et al., 2007). Despite its high sensitivity, no GST reconstruction based on
borehole thermal profiles is available for this part of the world. Instead, reconstructions of summer air
temperatures have been based on either tree-rings (e.g. Büntgen et al., 2006; Corona et al., 2010) or lake
sediments (e.g. Larocque-Tobler et al., 2010; Trachsel et al., 2010) for the last 500-1000 years, or both
(Trachsel et al., 2012). With rare exceptions (e.g. ice cores; Barbante et al., 2004), the other proxy data are
from sites at elevations that rarely exceed 2000m a.s.l. and all the other monitored permafrost boreholes in
Europe do not exceed 100 m of depth (see Harris et al., 2003). However, several papers describe GST
reconstructions for the last 500-1000 years using boreholes data at hemispheric or global scales (e.g.
Huang et al., 2000; Beltrami and Boulron, 2004).
The SSB data provides GST history from a high elevation site (3000 m a.s.l.). Such locations are important
because snow cover can affect significantly the GST (Zhang, 2005; Ling and Zhang, 2006; Cook et al., 2008).
They are also relevant with respect to glacier dynamics and their feedbacks with the global atmospheric
system (IPCC, 2013).
This paper reconstructs the ground surface temperatures inferred from this borehole and compares the
results with existing multiproxy reconstructions for the European Alps and elsewhere.

**2 STUDY AREA**
The Stelvio–Livrio area is a summer ski location, located between the Stelvio Pass (2758 m a.s.l.) and Mt
Livrio (3174 m a.s.l.), within the Stelvio National Park. The area is characterized by bedrock outcrops
(mainly dolostone), apart from some Holocene moraines (Figure 1a). The SSB borehole was drilled in 2009
and is only 10m from the PACE borehole, drilled in 1998 (46°30′59′′N; 10°28′35′′E, 3000 m a.s.l., Figure 1b).
Both boreholes are located on a flat  barren summit surface oriented  NNW-SSE. The side slopes (SSW and
NNE exposed) are gentle, the northern being only slightly steeper (14.1° vs 12.5° vs from the top down to
2900 m a.s.l.; Fig. 2, solid line). Despite their lithological homogeneity and their low porosity (< 5%, Berra,
personal communication), the two boreholes differ because in the PACE borehole two fractures filled by ice
were encountered at 42 and 90 m depth (Guglielmin et al., 2001) but no evidence of ice was observed
during the SSB drilling. Using PACE temperature profile and  typical thermal conductivity and heat flow
values cited in literature (4.0 Wm$^{-1}$K$^{-1}$, Clauser and Huenges, 1995; 85 mWm$^{-2}$, Cermak et al., 1992),
permafrost thickness in the SSSB borehole was estimated to be around 220 m.

**3 METHODS**
***3.1 Field data***
The SSB borehole was drilled in early July, 2010, using refrigerated compressed-air-flush drilling technology.
The stratigraphy was obtained by analyses of the cuttings (sampled every 10 m) and, for the first 100 m,
through analysis of TV logging.  Since September 2010, the thermal regime of the SSB borehole was
monitored with thermometers placed according to the PACE protocol (Harris et al., 2001). The accuracy of
the thermometers is 0.1°C and the resolution is 0.01°C.  The thermistors recorded the daily ground
temperature (minimum, maximum and average) at 20, 25, 35, 40, 60, 85,105,125,145,165,205,215 and 235
m of depth. Since 1998, the main climatic parameters at the site (air temperature, snow cover, incoming
radiation) have been monitored. Below the 20m depth, no significant seasonal variations in temperature
are recorded.
***3.2 Laboratory data***
The thermal properties of the three main facies observed in the stratigraphy were measured in the
laboratory at three different temperatures (0°C, -1°C; -3°C). Thermal diffusivity and specific heatwere
measured by NETZSCH Gerätebau GmbH (Selb, Germany) using a NETZSCH model 457 MicroFlashTM laser
flash diffusivity apparatus. Thermal diffusivity measurements were conducted in a dynamic helium
atmosphere at a flow rate of c. 100 ml/min between −3 °C and 0 °C. Specific heat capacity was measured
using the ratio method of ASTM-E 1461 (ASTM, 2003) with an accuracy of more than 5%. Density of the
rock at room temperature was determined using the buoyancy flotation method with an accuracy better
than 5%.  Thermal conductivity was calculated following Carlsaw and Jaeger (1959):
$$\lambda = \rho * c_p * \kappa,$$
where $\lambda$ is the thermal conductivity (W m$^{-1}$ K$^{-1}$), $\rho$ is the bulk density (gcm$^{-3}$), $c_p$ is the specific heat
capacity (Jg$^{-1}$ K$^{-1}$), and $\kappa$ is the thermal diffusivity (m$^2$ s$^{-1}$).
**3.3 Theory**
The temperature anomaly in the borehole at time t at depth z is modeled by the solution of the heat
equation

$$\frac{\partial A}{\partial t} - \frac{\partial}{\partial z}\left(\kappa \frac{\partial A}{\partial z}\right) = 0 \quad (1)$$

for the domain $(t, z) \in (-t_{max}, 0) \times (0, z_{max})$. Note that equation (1) can be derived from the classical
formulation of Carlsaw and Jaeger (1959) under the hypothesis that the density and the specific heat
capacity are constant with respect to the depth z (see also Liu and Zhang, 2014), which is a good
approximation for the SSB (see Section 4.1 and appendix). Further, we have indicated with $t_{max}$ the earliest
time for which we will reconstruct the GST and with $z_{max}$ the depth of the borehole. Equation (1) can be
solved to compute the temperature anomaly at any given past time t and depth z from the boundary values
A(t; 0) which represent the GST history. If the GST data A(t,0) are piece-wise constant, the solution of the
direct problem for equation (1) can be found explicitly (see Carlsaw and Jaeger, 1959). In our case, we need
to solve the inverse problem of finding the GST from the borehole data, which provide the anomaly
measured at present (t=0) or past times (t>0) at some depth z in the borehole.
In order to exploit the abovementioned explicit solution, it is customary to approximate the GST with a
piece-wise constant function (see Figure 3)

$$GST(t) = \begin{cases} \tau_k, & t \in [-t_k, -t_{k-1}] \\ \tau_\infty, & t \leftarrow t_N \end{cases} \quad (2)$$

where $t_k$, for k = 1,…,N, is the sequence of times in the past where we want to compute the value of the
GST, and the $\tau_k$'s are the unknown values to be computed. The diffusive nature of the heat equation has
the effect that fine details of GST signals are averaged away as time progresses. Therefore, in the field data,
one can find signals coming only from long wavelength GST variations occurred in the distant past,
whereas short wavelength signals are observable only if produced in the more recent history. In order to
take into account long and short wavelengths variations of GST where each of them makes sense, contrary
to the common use of choosing uniformly spaced time points, we choose

$$t_k = (1 + 0.2k)^2$$

so that the reconstruction points are closer to each other in the recent past and more separated for distant
ages. The choice of the parameter 0.2 is such that the reconstructed GST can contain signals of wavelength
of at least 33 years from 1600 onwards, 23 years from 1800 onwards, 16 years from 1915 onwards, 9 years
from 1985 onwards.
Once the sequence $t_k$ is chosen, the relation between the borehole temperature at depth $z_j$ predicted by
the model and the unknown values $\tau_k$ of the GST anomaly is linear. When comparing the anomaly A(z,t)
described by the above equation with the measured data in the borehole, one has to take into account that
measured data represent the superposition of the anomaly with a background signal (linearly increasing
with depth) coming from the heat flow and past climatic changes since the Last Glacial Maximum as found
for deeper boreholes by Safanda and Rajver (2001) or by Rath et al., (2012). This linear trend can be
identified by linearly fitting the data from the deepest part of the borehole (below 60m in our case).
Following (3), imposing that the borehole temperatures measured $T_j$ years ago at depth $z_j$ leads to a linear
system

$$L\vec{\tau} = \vec{m}, \quad (4)$$

where the column vector $\vec{\tau} = [\tau_1, \tau_2, …, \tau_N, \tau_\infty]$ collects the unknown GST values, $\vec{m}$ is the column vector
of detrended measured data and L is a matrix with M x (J + 1) entries (see the appendix). Each row in L (and
entry in the vector m), corresponds to a measured temperature in the well at present or at some time in
the past. In this fashion, the GST reconstruction can be based not only on a single temperature profile but
also on the variation of the temperature profile between the present and some years ago. To the best of
our knowledge, this possibility, which enhances the robustness of the reconstruction, has never been
exploited before in the literature. Given the detrended measures $\vec{m}$, we must compute the vector $\vec{\tau}$ solving
the linear system (4). However it is well known that the inverse problem for the heat equation (1) is severly
ill-posed and thus solving directly the linear system (4) would lead to a computed GST that would be highly
oscillating and very far from the true physical values for $\vec{\tau}$. It is then necessary to introduce a regularization
process by modifying the original problem (4), in order to obtain an approximation that is well posed and
less sensitive to errors in the right-hand-side of (4). Classical regularization techniques include the
truncated singular value decomposition (TSVD) and the Tikhonov regularization in standard form (Hansen,
1998), applied in Beltrami and Boulron, (2004) and Liu and Zhang, (2014), respectively. In this paper, we
propose the use of the generalized Tikhonov regularization, where the damping term is measured by a
proper seminorm. In practice, instead of dealing with the linear system (4), we solve the minimization
problem

$$\min_{\vec{\tau}} \|L\vec{\tau} - \vec{m}\| + \alpha \|R\vec{\tau}\| \quad (5)$$

where $\alpha > 0$ is the regularization parameter and R is the regularization matrix. The use of a regularization
matrix R for this application is a novelty although several other regularization smoothing parameters were
already used (i.e. Shen et al., 1992; Rath and Mottaghy, 2007) If R is simply the identity matrix, then the
problem (5) reduces to the standard Tikhonov method used in Liu and Zhang, (2014). When $\alpha$ is large the
restored GST is very smooth but the differences between the measured data and the temperatures in the
well that would be computed by (4) from the recovered GST are large. On the contrary, when α is too small
the data fitting is good but the GST becomes highly oscillating due to the ill-posedness. A good tradeoff is
not trivial and several strategies can be explored for estimating an optimal value of α: as an example, the
generalized cross validation (Golub et al., 1979) often provides good results.
A common choice for R is a finite difference discretization of a differential operator (Hansen, 1998]. In this
paper, we consider a standard discretization of the Laplacian so that the constant and linear components of
the solution are not damped in the Tikhonov regularization, (5) while we have a penalization of high
oscillations. The details of the chosen regularization and of the GST inversion employed are described in the
appendix.
3.4 Validation on synthetic data
In order to validate our GST inversion method we have generated a synthetic data set as follows. An ideal
GST was chosen (dashed curve in Fig. 4) and equation (1) was solved by a finite difference method with a
spatial grid spacing of 1 m. Homogeneous Neumann boundary conditions were imposed at the well bottom
and the ideal GST as Dirichlet data at z=0, thus obtaining synthetic data for the measurements of
temperature in the well. The computed temperatures were saved for the depth at which the real
thermometers in SBB are located (see Section 3.1), for the present time, as well as for 1, 2 and 3 years
before present. We then used the generated data as input to the inversion algorithm described in the
previous section and compared the reconstructed GST with the ideal one used to generate the synthetic
data.
In the first experiment we fed our inversion algorithms only with the synthetic data for the present time.
The value of alpha that best fits the exact GST is alpha=0.15, but in Figure 4 one can see that also varying
this value by 33% the reconstructed GST does not vary significantly.
Next we fed the inversion algorithm also with the synthetic data for the past years. First, the inversion is
expected to be more accurate since the algorithm can average not only on the temperature at a given
depth but also on the variation of the temperature in the last years at that depth. Moreover, the algorithm
should also be more robust, since it relies on a larger data set. Both these effects can be appreciated in Fig.
5, where it can be seen that the inversion in the last 50 years is more accurate than the inversion of Fig. 4
and that a wider variation in the value of alpha is possible without affecting very much the quality of the
reconstruction.

**4 RESULTS**
**4.1 Permafrost temperature, thermal properties and GST reconstruction**
The SSB stratigraphy is characterized by four different facies of dolostone (Figure 6):  a massive dolostone
(from grey to pinky grey) comprises more than 90% of the profile; three other facies (white dolostone,
black stratified limestone, brownish dolostone) are thin intercalations (maximum 3.5 meters of thickness
and located mainly in the first 42 m). In particular facies d, was not analysed for thermal analyses because is
very limited and it does not have any lateral continuity.
The mean annual thermal profiles of the last three years (2013-14-15) show a negative gradient between
20 m (a depth corresponding approximately to the depth of zero annual amplitude, ZAA) and 60 m that
does not vary (-0.8°C/100 m in all the three years). At greater depth, the gradient is positive with slightly
different slopes between 60-105; 105-125; 125-205; 205-215 and 215-235 (Figure 7 and Table 1).
Table 2 shows the thermal properties of the three main stratigraphic facies encountered in the borehole.
Facies a and c show  similar density and thermal properties while facies b has higher density and higher
conductivity. All facies have heat capacity values that increase with a decrease of  temperature. In facies a,
this behavior occurs also for thermal conductivity and  diffusivity values. In contrast, facies b and c show a
reversed bell shape behavior, with the minimum value recorded at -1°C and an absolute maximum at -3°C.
Therefore, from a thermal point of view, only facies b is different. Moreover, at depths below the level of
zero annual amplitude, this facies occurs only at depths of 34.5m and 90 m with a negligible thickness (2
and 1 m respectively) and at 142.5 m and 205 m where it reaches 3-3.5 m in thickness. Clearly, the thermal
influence of this facies is negligible,  indeed, the gradient between 60 and 235 m is approximately the same
as that   between 60 and 105 m and between 125 and 205 m. The effects of the different thermal
diffusivities measured in the different facies of the SSB borehole are also illustrated in Figure 8 where is
possible to notice that the difference of temperature *a posteriori* between a model with a constant
diffusivity equal to the average value of the facies a between 0°C and -1°C (red dots) and the model with
the different diffusivities for each different facies layer (blue dots) is absolutely negligible (< 0.02°C) in all
the depths with the exception of the uppermost (20 m) where the difference is  higher but still very low
(0.06°C).
According to the model proposed in the Methods, we found the best fitting with the thermal profiles
(Figure 7) using an heat flow of 70 mWm$^{-2}$ (Della Vedova et al., 1995), a thermal diffusivity value equal to
the mean between the value obtained for 0°C and -1°C for facies a,  which is the more widespread in the
borehole and an alpha value of 0.95 as shown in figure 9.
The linear system (4) was assembled including the detrended data measured at SSB in 2015 ($T_j = 0$, in
2014 ($T_j = 1$ and 2013 ($T_j = 2$, at the 13 depths listed in Section 3.1, resulting in 39 equations. The
anomalies of the GST reconstruction obtained with respect to the reference period between 1880 and 1960
has been computed using the value of $\alpha = 0.95$ for the regularization parameter (Figure 10).

## 5 DISCUSSION

### 5.2.1 GST and current air temperatures

In cryotic environments, snow cover can influence GST variability both in space and in time (e.g. Zhang,
2005; Schmidt et al., 2009; Morse et al., 2012; Rodder and Kneisel, 2012; Schmid et al., 2012; Guglielmin et
al., 2014). This is especially the case for alpine areas where  topography influences both the re-distribution
of the snow by wind-drift and actual snow cover evolution (e.g. melting date and duration**).  Nevertheless,
GST and air temperature are well correlated ($R^2$ = 0.8027) and present a very similar pattern over the last
15 years with only a slight warming (Figure 11).  This relatively slight effect of snow at this site is probably
due to the high wind velocities during winter that, on average, prevent buildup of a  thick snowpack. Figure
12 illustrates the temporal variability of  snow cover on the GST.  In general, the highest (>±5°C) differences
between mean daily GST and  mean daily air temperature occur when there are large drops of air
temperature during the winter.  Sometimes, large differences occur also when there are large drops of air
temperature during the summer where there is little or no snow cover, because of  high solar radiation that
heats the ground surface. Correlation is even better between monthly mean air temperature, mean annual
air temperature (MAAT) and  mean annual ground surface temperature (MAGST) ($R^2$ = 0.8712 for this
latter).  This agrees with the results of Zhang and Stamnes, (1998) who found that, in a flat area in northern
Alaska, changes in  seasonal snow cover had a smaller effect than MAAT on the ground thermal regime.

### 5.2.2. GST Fluctuations between 1950 and today

Our reconstruction after the cold GST anomaly, between 1906 and 1941 AD, shows a slightly positive peak
(ca. 0.1°C) in 1930 and afterwards a very unstable period with a first sharp decrease of temperature until
1989 (between -0.2 and -0.6°C) and a second even sharper increase, reaching in 2011 the uppermost GST
anomaly value of the last 500 years (around 1°C).
On a regional scale, the Stelvio data can be compared with the MAAT obtained for the Alps by Christiansen
and Ljungqvist, (2011) (Figure 10) and Trachsel et al., (2010). The maximum of the slight temperature
increase  during the first half of the XX century in the Stelvio data (1930)  falls exactly in the middle of the
relative warming period between 1925 and 1935 in the Alps found by Trachsel et al., (2010) and is in good
agreement with the date (1928) indicated by Christiansen and Ljungqvist, (2011). Later, the sharp GST
anomaly decrease was delayed in the Stelvio data (1989) with respect to 1950-1965 period found by
Trachsel et al., (2010) and 1965-1975 period found by Christiansen and Ljungqvist, (2011). Finally, the most
recent increase of temperature culminated in the Alps in 1994 (Christiansen and Ljungqvist, 2011) while in
the Stelvio data at 2011.

### 5.2.3 The Little Ice Age (LIA)

The Stelvio reconstruction shows a long period of negative anomaly between 1560 and 1860 AD with the
colder conditions (< -2*S.D.) between 1683 and 1784 AD with a peak of -1.5°C around 1730 AD.  This period
of negative anomaly falls within this well-known cooling period (LIA). It is recognized in several kinds of
proxy data  although there are differences both in magnitude and in timing across the world. According to
Neukom et al., (2014), synchronous cold temperature anomalies occurred at decadal scale in both
hemispheres between 1594 and 1677 AD. They also found two phases of extreme cold temperature in the
Northern Hemisphere with the first between 1570 and 1720 AD and the second between 1810 and 1855.
Syntheses of the LIA in the European Alps have been presented by Trachsel et al., (2012) and Christiansen
and Ljungqvist, (2011). Considering the common colder periods in these two Alpine syntheses, the LIA has
three main negative peaks at 1570-1600; 1685-1700 and 1790-1820 AD.
The LIA period has been also characterized by a widespread worldwide glacier advance, although the
comparison between glacial evidences and temperature fluctuations are problematic because glaciers
respond with different time scales (mainly depending  on their size) and reflect also the precipitation
regime,  which is even more variable in space and time. According to Holzhauser et al., (2005), the LIA
advance  of the main Swiss Glaciers has three peaks around respectively 1350, 1640 and 1820-50 AD with
the two later phases almost synchronous also in the Eastern Alps (Nicolussi and Patzelt, 2000).
Close to the location of the Stelvio borehole, the maximum LIA advance was diachronous. Nearby glaciers
show a maximum LIA advance in 1580 AD (Trafoi Valley glacier; Cardassi, 1995), around 1770 AD (Solda
Glacier; Arzuffi and Pelfini, 2001) and in 1600 AD (La Mare Glacier; Carturan et al., 2014).
The borehole area was presumably overcapped by the Vedretta Piana Glacier until 1868. Due to the
geomorphological position (on a watershed divide) the possible glacier should have been very thin and
possibly cold based, as already stressed by Guglielmin et al., (2001). On the other hand, considering figure
10, the glacier should have been present in the borehole area with a buffering effect only between 1711
and 1834 AD, with a peak at 1760, when the difference between the GST anomaly and the MAAT anomaly
was maximum. This peak is pretty similar to the peak of the LIA in the Solda Glacier (1770 AD) but not to
the peak in the Trafoi glacier (1580 AD); this could be related to Vedretta Piana having a more similar
glacier size and aspect (NE-N)  to the Solda Glacier than to the Trafoi Glacier, although this latter is the
closest to the Vedretta Piana.
**5.2.4  Other permafrost borehole temperature reconstructions**
Several deep Alaskan boreholes have been used to demonstrate the XX century warming (e.g. Lachenbruch
and Marshall, 1986; Lachenbruch et al., 1988) but only a few studies in Europe illustrate GST
reconstructions that span a time period greater than 100-150 years (e,g, Isaksen et al., 2001, Guglielmin,
2004). In North America, only Chouinard et al., (2013) shows GST pattern of the last 300 years in the
context of the permafrost of Northern Quebec. There, after the LIA (1500-1800 AD), it was found an almost
constant and marked warming of ca 1.4 °C until 1940, followed by a cooling episode (≈0.4 °C) which lasted
40–50 yr, and finally a sharp ≈1.7 °C warming over the past 15 yr.
There is a some similarity between the Stelvio reconstruction and the pattern of Canadian permafrost GST
reported by Chouinard et al., (2013) after the LIA. Indeed, also in our site there was an almost simultaneous
but greater cooling (0.9°C) in the period between 1941 and 1989, followed by a sharp warming of ca 1.7°C.
On the other hand, GST reconstructions can be  obtained with different models and it is interesting to
compare our data with, for example, the PMIP3/CMIP5 simulations that include the effect of aerosol
forcing by Garcia-Garcia et al., (2016): there, in the last 500 years, the GST shows a cold anomaly (LIA)
between 1582 and 1840, with the most negative peaks between 1798 and 1840, slightly delayed with
respect to our data.

## 5 CONCLUSIONS

The general climatic pattern of the last 500 years recorded by this mountain permafrost borehole is similar to the majority of other studies in the European Alps and Central Europe. The main difference concerns post LIA events. In fact, the different multidisciplinary proxies considered (see Figure 13) do not indicate cooling between 1940 and 1989, with the exceptions of the shorter and less severe cooling found for the Alps. It is also relevant to stress that the rapid and abrupt GST warming (more than 0.8°C per decade) recorded between 1990 and 2011 in the Stelvio borehole data is similar to the warming recorded in permafrost in northern Quebec. This warming trend is of the same magnitude as the increase of MAAT between 1990 and 2000 in Central Europe (Dobrovlny et al., (2010), and is approximately double that found for the MAAT in the Alps and for Europe as a the whole (Luterbacher et al., 2004).

The Stelvio borehole ground surface temperature reconstruction also allows one to estimate changes in the Vedretta Piana glacier. This glacier presumably buried the site of the Stelvio borehole with an ice thickness sufficient to exert a significant buffering effect upon the ground thermal regime between 1711 and 1834 AD. This was a time when the difference between the Stelvio GST anomaly and the MAAT anomaly was greatest.

328

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

## 7 ACKNOWLEDGEMENTS

The SSB borehole was drilled and equipped thanks to the Project "Share Stelvio" managed by EvK2-CNR and
funded by Regione Lombardia. The research was also funded through the PRIN 2008 project "Permafrost e
piccoli ghiacciai alpini come elementi chiave della gestione delle risorse idriche in relazione al Cambiamento
Climatico" leaded by Prof. C. Smiraglia. Special thanks to the Stelvio National Park, SIFAS and Umberto
Capitani for the permissions and the logistical support. We want also to thank you very much Prof. Hugh M.
French for the revision and the English editing of the manuscript.

## Figure and Table Captions

Figure 1. Study area: (a) Location of the study area with the surrounding glaciers and the reconstructed
glaciers limits of the area (VPG = Vedretta Piana Glacier; TFG = Trafoi Glacier; SG = Solda Glacier; LMG = La
Mare Glacier; PACE = Pace Borehole; SSB = Share Stelvio Borehole; (b) View of the drilling equipment during
the realization of the SSB borehole in summer 2009.
Figure 2. Topography of the SSB site: a) Digital Elevation Model (5 m resolution) of the SSB site and b) SSW-
NNE (solid line) and N-S (dashed line) transects through the Stelvio summit. Horizontal and vertical scales
as well as thermistor chain position and depths are plotted to the same scale.
Figure 3. Example of a GST history parametrized by equation (2).
Figure 4. Synthetic data for the present time. It is remarkable that also varying the alpha value by 33% the
reconstructed GST does not vary significantly. Legend: 0.1 = blue line; 0.15 = green line; 0.2 = orange line.
Figure 5. Synthetic data for three past years (2013, 2014 and 2015). It can be seen that the inversion in the
last 50 years is more accurate than the inversion of Fig 4. Legend: 0.15 = green line; 0.2 = orange line; 0.25 =
red line.
Figure 6. Share Stelvio Borehole (SSB) Stratigraphy. Legend:  (A) facies a (massive dolostone from grey to
pinky grey); (B) facies b (white dolostone); (C) facies c (black stratified limestone); (D) facies d (light brown
dolostone).
Figure 7. SSB mean annual ground temperature profiles on 2013, 2014 and 2015.
Figure 8 Effects of different thermal diffusivity used in the model. The temperature profiles *a posteriori* of
2015 obtained in the case of a constant thermal diffusivity value of the more widespread facies (a) (red
dots) and in the case of with a multilayers thermal diffusivities according the different facies according Fig.
6 (blue dots). The bars indicate the variations of the measured temperature in the same year.
Figure 9 Example of different GST history with different alpha with the extreme of heat flow values known
for the region. Legend: green lines are obtained with a heat flow of 70 mWm$^{-2}$ while red lines with 85
mWm$^{-2}$ . The different symbols indicate different alpha value (0.95 = solid line; 1.1 = empty dots; 0.8 =
triangles).
Figure 10. Comparison between the anomaly of the mean annual GST reconstructed by SSB borehole (black
thick line), its uncertainty range (red shaded) and MAAT anomaly reconstructed for the European Alps by
Christiansen    and    Ljungqvist    (2011)    (grey    line    with    dots;    data    available    online    at:
ttps://www.ncdc.noaa.gov/paleo/study/12355) both respect the same reference period (1880-1960).
Figure 11. Trend of monthly mean of GST (red line) and Air temperature (blue line) at SSB since 1998. The
red and blue dashes lines are respectively the linear regression of the GST and Air temperature.
Figure 12. Effect of the snow cover at SSB. The winter 2010/11 is representative of the average conditions
of the snow cover at SSB while the following season 2011/12 was the snowiest of the whole monitoring
period. The difference between the daily mean GST and air temperature (ΔGSTair; black line) shows the
greater values during the greater drop of the air temperature (green line) during the winter due to the
insulating effect of the snow cover (blue line) whereas the few episodes of high ΔGSTair in the summer are
may due to the solar radiation that warms up the ground surface.
Figure 13. Main climatic events enhanced by anomalies of MAAT through different proxy in all Europe: A,
(modified from Luterbacher et al., 2004); Central Europe: B, (rielaborated from Dobrovolný et al., 2010;
Alps: C, (modified from the same data of Figure 5, Christiansen and Ljungqvist, 2011) and SSB: D, (this
paper).
Figure 14. Comparison of predictions of the forward model for the same GST and different geometrical
setups. Legend: 1D = red dots; 2D flat terrain = blue line; 2D N-S green line; 2D SSW-NNE orange line; 3D
dashed black line. (See the Appendix for the details).
Table 1. Thermal gradients (°Cm-1) on 2013; 2014 and 2015 in the different depth intervals of the profile
below the zero-annual amplitude that is approximately at 20 m of depth.
Table 2. Thermal properties of the three different facies occurred in SSB borehole measured in the
Laboratory at three different steps of temperature (0; -1 and -2°C).
Table 1

|  | 20-60 m (°Cm$^{-1}$) | 60-105 m (°Cm$^{-1}$) | 105-125 m (°Cm$^{-1}$) | 125-205 m (°Cm$^{-1}$) | 205-215 m (°Cm$^{-1}$) | 215-235 m (°Cm$^{-1}$) | 60-235 m (°Cm$^{-1}$) |
|---|---|---|---|---|---|---|---|
| 2013 | 0.0088 | -0.0072 | -0.0048 | -0.0075 | -0.0128 | -0.0058 | -0.0072 |
| 2014 |  |  | -0.0046 | -0.0074 | -0.0128 | -0.0056 |  |
| 2015 | 0.0086 | -0.0077 | -0.0045 | -0.0073 | -0.0128 | -0.0055 | -0.0072 |


Table 2

|  | Density (gcm$^{-3}$) | Diffusivity (10$^{-6}$m$^2$s$^{-1}$) | Heat capacity (Jg$^{-1}$K$^{-1}$) | Conductivity (Wm$^{-1}$K$^{-1}$) |
|---|---|---|---|---|
| **Facies a** | 2.7 |  |  |  |
| 0°C |  | 2.2 | 0.7 | 4.5 |
| -1°C |  | 2.1 | 0.8 | 4.4 |
| -3°C |  | 2.1 | 0.8 | 4.4 |
| **Facies b** | 2.8 |  |  |  |
| 0°C |  | 2.8 | 0.8 | 6.2 |
| -1°C |  | 2.8 | 0.8 | 6.2 |
| -3°C |  | 2.8 | 0.8 | 6.2 |
| **Facies c** | 2.7 |  |  |  |
| 0°C |  | 2.0 | 0.8 | 4.0 |
| -1°C |  | 1.9 | 0.8 | 3.9 |
| -3°C |  | 1.9 | 0.8 | 4.0 |


## Appendix 1: Details of the regularization and inversion technique

The temperature anomaly in the borehole at time t at depth z is modeled by the solution of the heat
equation

$$\frac{\partial A}{\partial t} - \frac{\partial}{\partial z}\left(\kappa \frac{\partial A}{\partial z}\right) = 0 \quad (1)$$

for the domain $(t,z) \in (-t_{max}, 0) \times (0, z_{max})$. If the boundary data A(t,0) is piece-wise constant, then
the solution of the direct problem for equation (1) can be found explicitly (see Carlsaw and Jaeger, 1959). In
fact, the anomaly observed in the borehole t years ago, originating from a GST that has been constant
except for an increase of $\delta$ °C between $t_2$ and $t_1$ years ago is:

$$A(t,z) = \delta\left[erfc\left(\frac{z}{\sqrt{4\kappa(t_2 - t)}}\right) - erfc\left(\frac{z}{\sqrt{4\kappa(t_1 - t)}}\right)\right]$$

The above formula of course makes sense only for $t < t_1$ and the value $t = 0$ corresponds to present time.
For the purpose of reconstructing the GST history, it is customary to approximate it with a piece-wise
constant function (see Figure 3)

$$GST(t) = \begin{cases} \tau_k, & t \in [-t_k, -t_{k-1}] \\ \tau_\infty, & t \leftarrow t_N \end{cases} \quad (2)$$

where $t_k$, for k = 1,...,N, is the sequence of times in the past where we want to compute the value of the
GST, and the $\tau_k$'s are the unknown values to be computed. The prediction of model (1) for the borehole
temperature t  years ago, originating from the GST (2) is

$$A(z,t) = \tau_1 \varphi(z, t_1 - t) + \sum_{k=1}^{N} \tau_k[\varphi(z, t_{k+1} - t) - \varphi(z, t_k - t)] - \tau_\infty \varphi(z, t_N - t), \quad (3)$$

where $\varphi(z,t) = erfc\left(\frac{z}{\sqrt{4\kappa t}}\right)$. Note that, once the sequence $t_k$ is chosen, the relation between the borehole
temperature at depth $z_j$ predicted by the model and the unknown values $\tau_k$ of the GST anomaly is thus
linear. The matrix L of the linear system (4) in the main text is thus

$$L_{j,1} = \varphi(Z_j, t_1 - T_j)$$
$$L_{j,k} = \varphi(Z_j, t_{k+1} - T_j) - \varphi(Z_j, t_k - T_j)$$
$$L_{j,N+1} = \varphi(Z_j, t_N - T_j).$$

We point out that each row of the matrix L can have a different value of $T_j$, so that the GST reconstruction
can be based not only on a single temperature profile, but also on the variation of the temperature profile
between the present and some years ago. Further, it is not needed that the reconstruction times $t_k$ are
equally spaced in the past.
Given the detrended measures $\vec{m}$, we must compute the vector $\vec{\tau}$ solving the linear system (4). Note that
the inverse problem for the heat equation (1) is well-known to be severely ill-posed, the matrix L is strongly
ill-conditioned and its singular values decay exponentially to zero, with related singular vectors largely
intersecting the subspace of high frequencies (Serra-Capizzano, 2004). Therefore, since the right-hand side
$\vec{m}$ is affected by error measurements, solving directly the linear system (4) would lead to a computed GST
that would be highly oscillating and very far from the true physical values for $\vec{\tau}$. It is then necessary to
introduce a regularization process by modifying the original problem (4), in order to obtain an
approximation that is well posed and less sensitive to errors in the right-hand-side of (4). The Tikhonov
regularization usually provides better restorations than the trucated SVD, because it is characterized by a
smooth transition in the filtering of the frequencies and the smoothness of the transition can be somehow
chosen by manipulating the regularization parameter of the method (Hansen, 1998). In this paper, we thus
propose the use of the generalized Tikhonov regularization, where the damping term is measured by a
proper seminorm. In practice, instead of dealing with the linear system (4), we solve the minimization
problem

$$\min_{\vec{\tau}}\|L\vec{\tau} - \vec{m}\| + \alpha\|R\vec{\tau}\| \quad (5)$$

where $\alpha > 0$ is the regularization parameter and R is the regularization matrix. The presence of the matrix
R in (5) allows to impose some a-priori information on the true solution. Indeed, when minimizing (5), the
components of the solution belonging to $ker(R) = \{\vec{x} \, s.t. \, R\vec{x} = \vec{0}\}$ are perfectly reconstructed. In fact, if a
vector x belongs to ker(R) then ||Rx||=0 and hence the penalization term disappears in the minimization
problem (5) and consequently the data are perfectly fitted. Note that in order to guarantee the uniqueness
of the solution (5), the condition $ker(L) \cap ker(R) = \vec{0}$ has to hold.
In this paper, we use as regularizer a standard discretization of the Laplacian

$$
R = \begin{pmatrix}
-1 \\
2 \\
-1 & -1 & 2 & -1 \\
\ddots & -1 & 2 & -1 \\
& & & & \ddots & \ddots
\end{pmatrix}
$$

of size $(N-2) \times N$ and hence the constant and linear components of the solution are not damped in the
Tikhonov regularization (5).

## Appendix 2: comparison of 1D and higher dimensional models for SSB

In order to ascertain the effect of the terrain geometry we conducted a number of forward simulations
with the model (1) using as boundary data the synthetic GST shown in Fig 4 (dashed line) and already
employed for the sensitivity analysis.
First we computed the solution of the one-dimensional model (1). Next we computed the solution of the
corresponding three-dimensional model in a computational domain of 400X400 m centred around SBB and
500 m deep, whose the top surface was obtained from a DEM (with a resolution of 10m . Such domain was
discretized with the GMSH program and the heat equation was solved using linear Lagrange finite elements
in space and backward Euler in time. The mesh was refined until numerical convergence was observed and
in Figure 14 we present the results for a mesh with 1.3 million of tetrehedra. The numerical simulations
were performed with the HPC cluster of the Dipartimento di Matematica of the Università di Torino.
Figure 14 compares the temperature anomalies that each of the models would predict at SSB at present
time. The red dots are the predicted well anomalies at the depth of the thermometers at SSB.  One can see
that the predictions of the two-dimensional model with flat terrain (blue line) almost coincide with those of
the one-dimensional one. Furthermore, the two-dimensional model applied to the section with the steeper
sides (the SSW-NNE one, orange line) gives rise to predictions that are within the instrumental error
(±0.1°C) whereas the N-S section (red line), which has a flatter terrain, gives rise to predictions that are
quite close to those of the one-dimensional model. The predictions of the 3D model (dashed black line) are
very close to the 2D flat and the 2D N-S (with difference always < 0.03°C).
Finally, let us remark that for the forward model, a numerical 3D simulation takes hours to complete on 16
computing nodes of our HPC cluster. Using a numerical multi-dimensional simulator in the inverse problem
would of course require to compute several times the forward model and would thus take a lot longer than
the few seconds in which our proposed method can compute the reconstructed GST depicted in Fig. 10.












Fig. 1

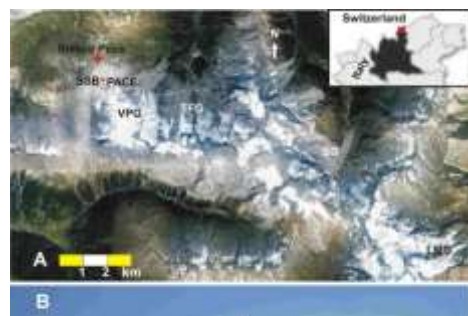

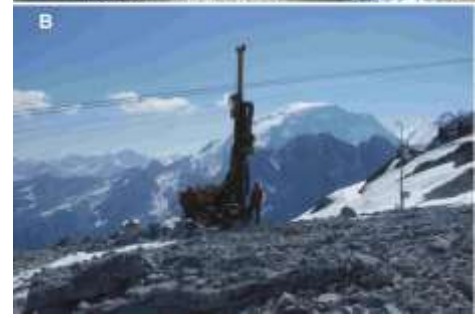














Fig. 2

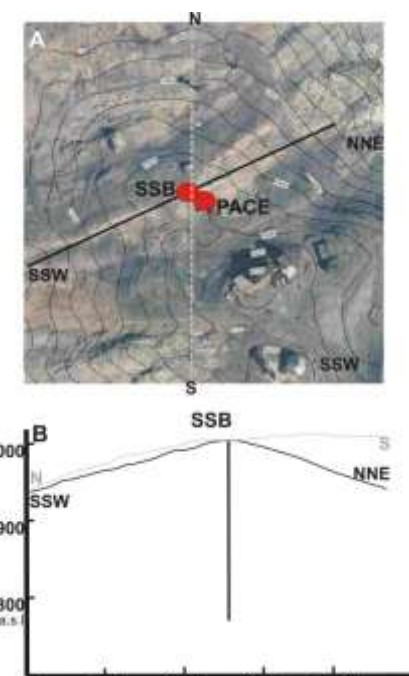




Fig.3

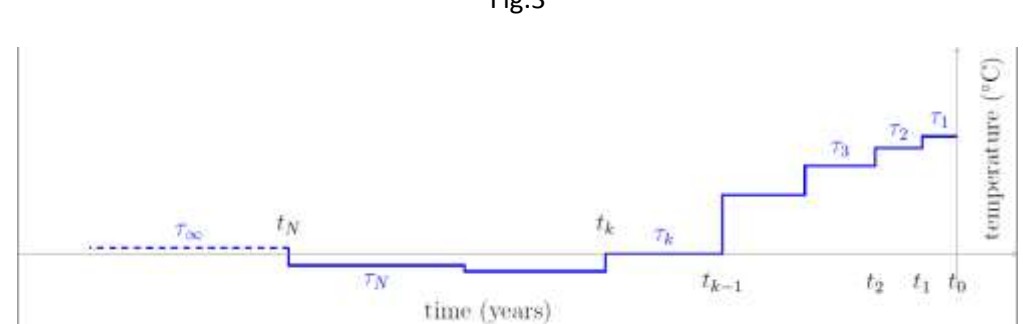











Fig.4

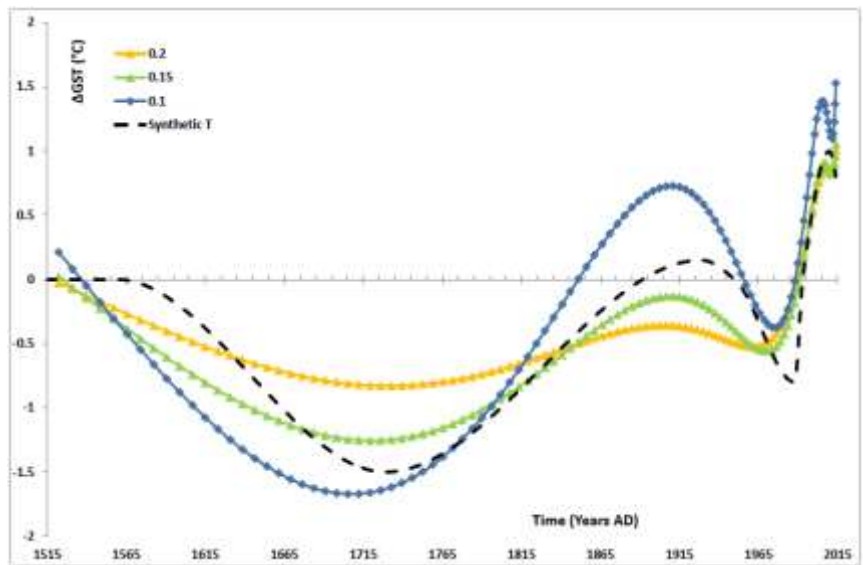



Fig. 5

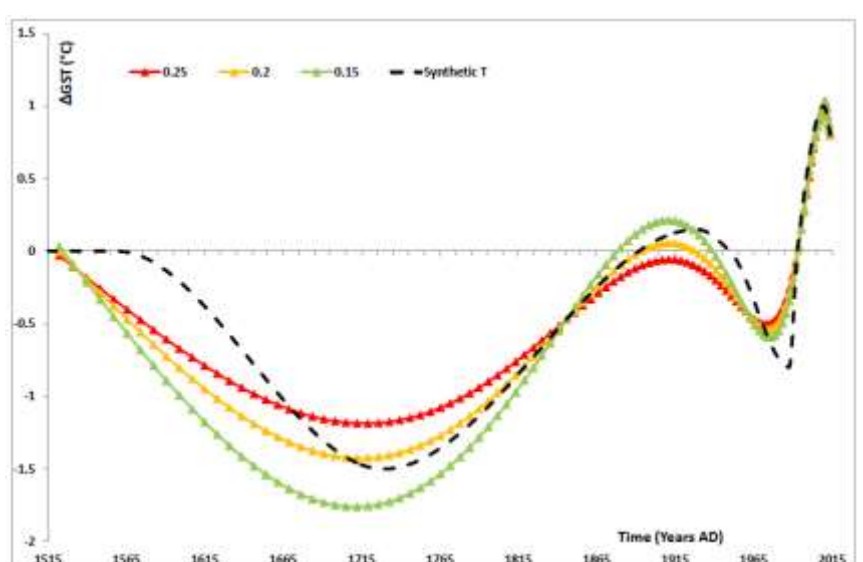











Fig. 6

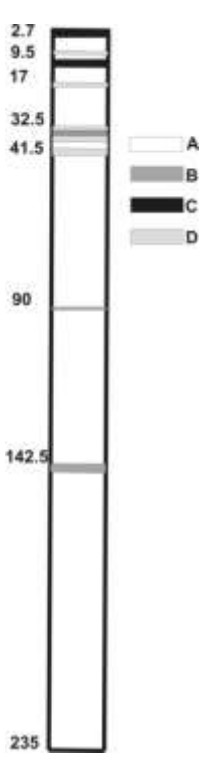


Fig. 7

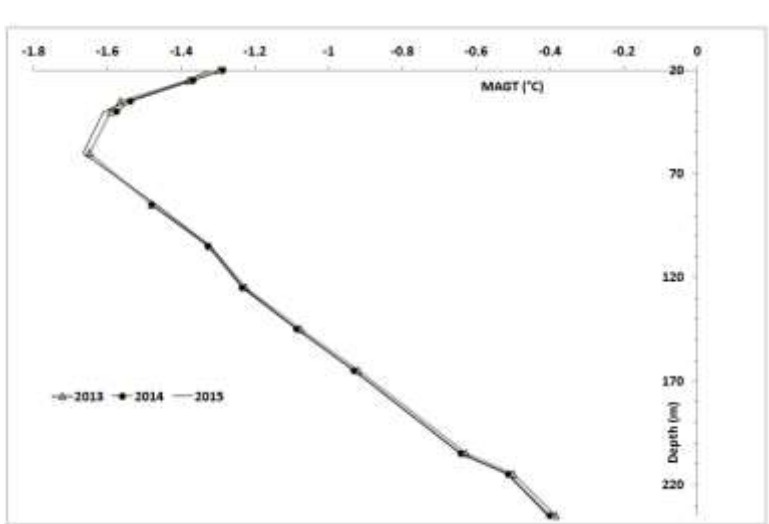









Fig. 8

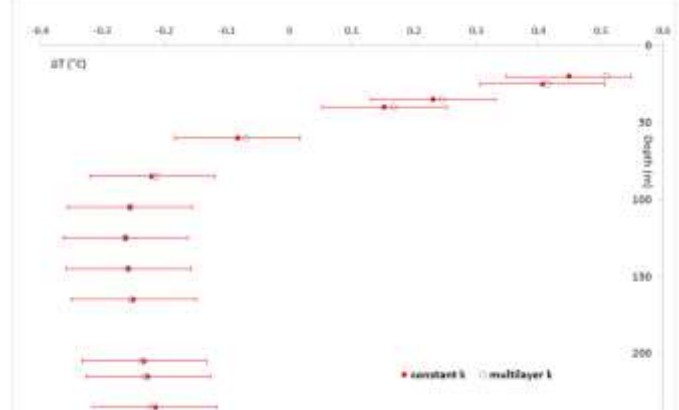



Fig. 9


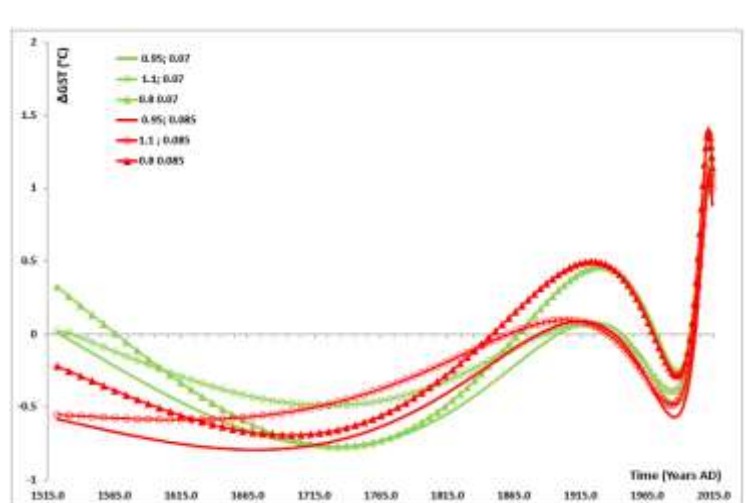










Fig. 10




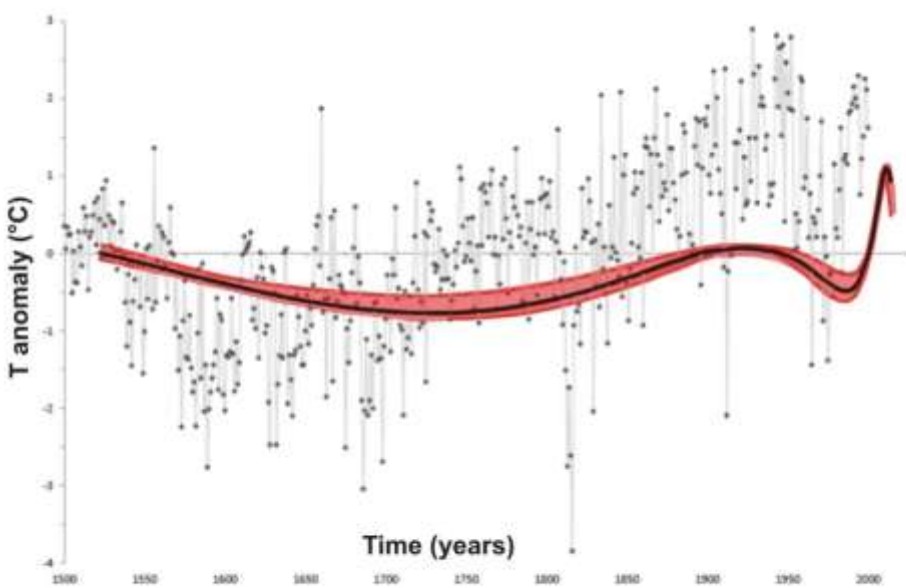



Fig. 11


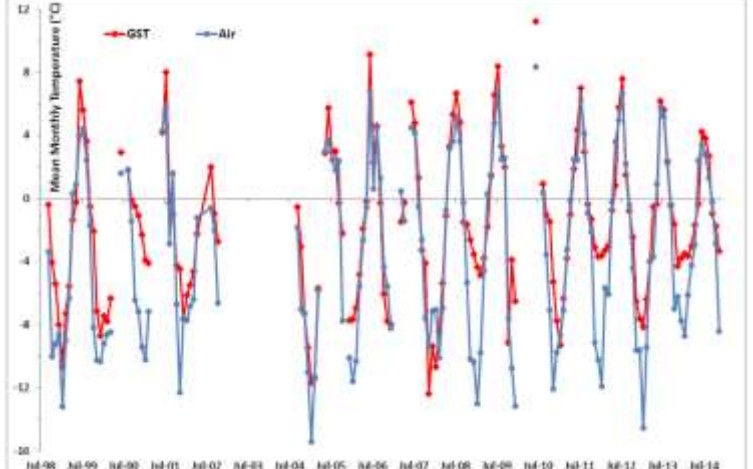



Fig. 12


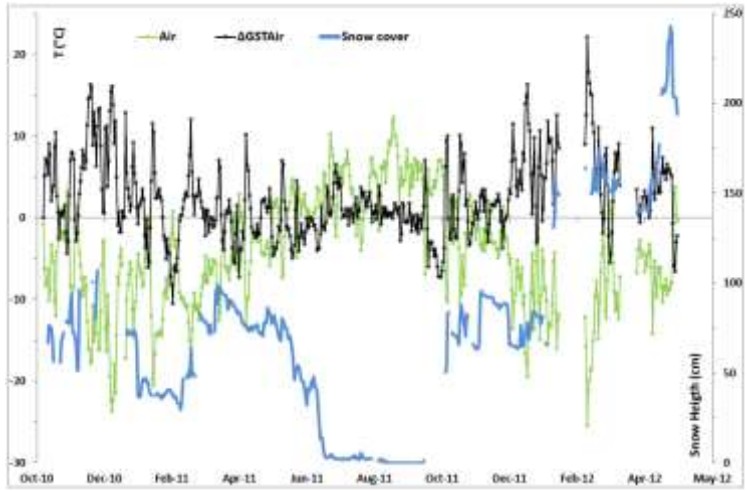


Fig. 13

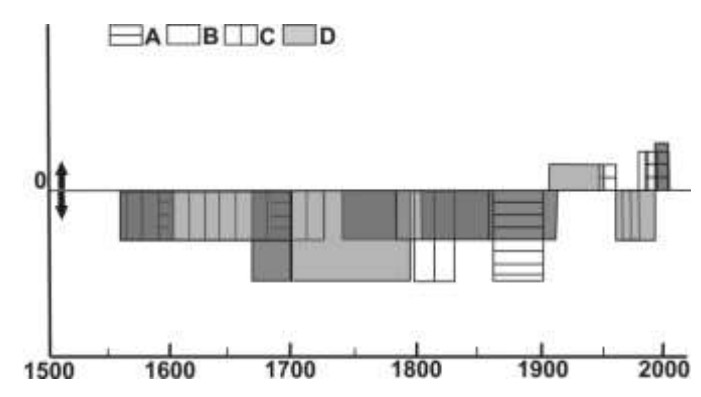


Fig. 14

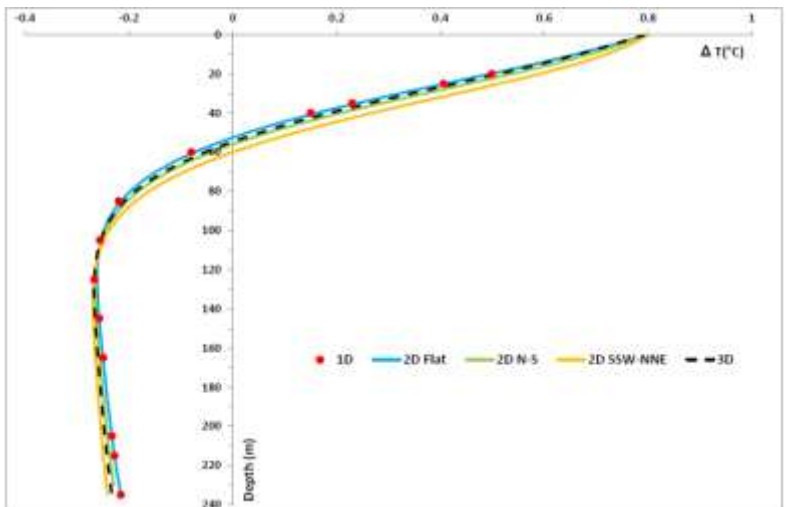



