# Peer review of "GROUND SURFACE TEMPERATURE RECONSTRUCTION FOR THE LAST 500 YEARS OBTAINED FROM PERMAFROST TEMPERATURES OBSERVED IN THE STELVIO SHARE BOREHOLE, ITALIAN"

_Climate of the Past, 2017_

## Referee Comment (RC1) · Anonymous Referee #1 · 26 Apr 2017

In summary, the manuscript by M. Guglielmin et al. gives results of reconstructing the ground surface temperature (GST) history by inversion of data from a borehole in the Italian Alps. Generally, the data presented is very interesting and valuable and a study about GST history using this data contributes to the knowledge about the interaction of climate changes and permafrost.

However, I do have some concerns regarding the procedure and method presented here – particular regarding uncertainty: the manuscript provides a too "straightforward" GST history reconstruction, neglecting any uncertainty ranges in the parameters involved. What about different regularization parameters? Could synthetic temperature profiles and the corresponding reconstruction of GST history give more insight into the method? Some additional (sensitivity) studies and a corresponding critical discussion is necessary, in my opinion. Data is available only at a few depth levels (Figure 5). How about the minimum value at 60 m particular? What would be the influence on the result if possible variations of the data beneath and above this depths occur? Regarding the result in Figure 6, it is not possible to assess any uncertainties or to distinguish between effects arising from the "smoothed" data and the presented inversion.

As a consequence, I consider numerical methods for reconstructing GST history in this case (mountain area, unfrozen water content, uncertainties...) superior to analytical methods. If the latter ones are applied, a thorough justification and critical discussion must be given. Regarding this, presenting only one GST history result in Figure 6 and using this for the interpretation does not comply with the demands/conclusions of the manuscript.

Other comments:

- Abstract: a significant part of the manuscript deals with the method used for GST reconstruction. Therefore, the method etc. should be mentioned in the abstract. Also, some important information one the borehole, such as depth, temperature ranges etc.

- Line 18: ...roughly double the MAAT...is not clear, doubling of the increase of MAAT?

- Line 22: linearly only, if there is additionally no heat production.

- Lines 27/28: the propagation of signals is a diffusive process, therefore it is interesting where a maximum of a signal occurs.

- Line 13: no groundwater flow only within continuous permafrost, more explanation needed.

- Line 41: the authors should justify this statement by some calculation ("...for much of the last millenium"), due to the diffusive nature, the signal of the last millennium is

not only visible in the upper 230 m, so a "truncation" of this signal may lead to a bias. - Line 74: heat flow in mountainous areas may differ stron gly from the typical literature values, so an estimation of permafrost thickness only using this value is questionable - Line 87, laboratory data: there is no information about porosity, this parameter is important with respect to the latent heat effect. It is only mentioned that there is no evidence of ice encountered, although ice has been encountered in a very close borehole only a few meters apart. The temperature range (-2-0 °C) is within the very range where a coexistence of both phases in soil/rock occurs (see references below).

- Line 93: if the accuracy of the measurements of thermal properties is around 5%, it is then necessary to state values in table 2 accordingly (three decimal places are certainly not applicable). The same applies to Table 1 regarding the temperature gradients.

- Please check generally, if "°C" is used for absolute temperatures and "K" for temperature differences, this makes the distinction easier.

- Line 126: the linear trend in the "deepest par" (below 60 m) can be still disturbed by a transient signal from the surface, so I does not really represent a background signal. This should be discussed.

- Line 139: 0.2 is choses, why? What would be the effect different values? - Line 147: it enhances the robustness: can this be justified?

- Line 167: How is the optimal parameter alpha determined? What is the influence on the results for different values of alpha? The regularization has been applied in earlier works (see references below).

- Line 208: what are the 13 depths listed in section 3.1? A figure would be helpful.

- Figure 1 A and Figure 2: Scale is missing.

- Figure 3: labels are missing (time/temperature).

- Figure 6: vertical axis is Delta T, referring to what?

- Figure 7: What is 0.02? Is the linear regression really justified? How about correlation coefficients?

References: Mottaghy, D. & Rath, V. (2006), Latent heat effects in subsurface heat transport modelling and their impact on palaeotemperature reconstructions, Geophysical Jounal International 164, 236-245.

Romanovsky, V. E. and Osterkamp, T. E. (2000), Effects of unfrozen water on heat and mass transport processes in the active layer and permafrost. Permafrost Periglac. Process., 11: 219–239.

Rath, V. & Mottaghy, D. (2007), Smooth inversion for ground surface temperature histories: estimating the optimum regularization parameter by generalised cross-validation, Geophysical Journal International 171 (3), 1440-1448.

---

## Referee Comment (RC2) · V. Rath (Referee) · 10 Jul 2017

While the manuscript contains interesting data worth while publishing, I'm highly skeptical with regard to the particular approach taken by the authors. It leaves open several important questions, cannot be regarded as really novel, and is not well described. Therefore I think the article could only be published in CP after very thorough (major) revisions.

[Figure]

**Major comments:**

**P2, L43-55:**These paragraphs read as f the existence of permafrost is essential to the reconstruction of past surface temperatures from borehole temperature profiles, which is not correct. Furthermore the early work of Lachenbruch  Marshall did not take into account the generic permafrost-related processes as freezing/thawing. This is related to the question of the existence of significant porosity (not even mentioned in the text). If water/ice-filled porosity is very small, the "dynamic" effects of permafrost are of course negligible. But then, the title may be a misnomer, and permafrost should be omitted there ("subsurface temperatures" instead of "permafrost temperatures").

**P2, L53-53:** I disagree with the sentence regarding the importance of topographic effects. Even on a fully symmetric mountain these effects will be present. Moreover, the differences in insolation will produce an asymmetric regime. A N-S slice thus would have been more relevant in Fig. 2, as this would be more characteristic with respect to the surface temperatures, and also would show more asymmetry. In order to be published, there should be a quantitiative assessment of the topographic effects with respect to the 1-D model used for inversion.

**P3, L83:** Please comment on these literature values.

**P4, 3.2 Lab data:** In table 2 there are three facies with quite different properties. How did you use this in your inversion? Note that a correct layered solution is given in Bodri & Cermak 1995.

**P4ff, 3.3 Theory:** While the authors try do give a description of their approach, there are many claims or assumptions which need clarification. As the theory section is already rather long, it might be useful to put the details into an appendix, and concentrate on the essentials in the main text.

- Any assumptions on the physical limitations of the model should be mentioned in the text, e.g. porosity, latent heat release, properties regarded constant.

- Why and how detrend? Is this detrending unique? (L132ff). This trend surely does not only realut from regional heat flow density, bur includes all earlier events, which often leads to an approximately linear signal at these shallow depths (amongst others, Safanda & Rajver 2001, Rath et al. 2012). I would also not call it a detrending - it is a different nontrivial inverse problem for background heat flux. This is also related to the choice of the length of the temperature history to be reconstructed and the relevance of the $\tau_\infty$ resulting from the inversion.

- Time lapse used in this inversion(L152-154)?

- This sentence is not comprehensible (L156-157).

- Why is Tikhonov better than TSVD? A comparison figure would help. (L 164ff).

- Which method was used to choose alpha for this study (L169ff)?

- Smoothing regularizations have been used many times in the past (L175) - see Shen et al. 1992, Bodri & Cermak 2007, also the references given by Referee #1.

- This sentence is incomprehensible (L178-179).

- **R** is not square. Which solver is used? Boundary conditions in **R**? Is the condition of L180 fulfilled with your construction of **L** and choice of **R**? For **R** to be a discrete approximation to a differential operator a factor of $(\Delta t)^{-2}$ is required (for constant $\Delta t$ in L187.

**General:**

An assessment of uncertainty is missing, which is absolutely necessary particularly in the case of an ill posed problem. This is even more important when using a simple 1-D model which neglects so many effects which may bias the results. While not improving with respect to the mentioned physical assumptions, already Monte Carlo studies

and sensitivity considerations would help in this respect. Any result and interpretation based on a single inversion can not be considered as reliable and is prone to bias. The lack of a critical evaluation, which is essential when reconstructing ground surface temperature histories, makes it difficult to judge the value of the results obtained.

**Minor items:**

- Use dots, not commas for the decimal in the tables.
- Caption figure 3 (most important result of the study) should be informative.
- Facies d in Figure 4 not in table 2.
- Marking facies (Figure 4) and paleoclimatic evens (Figure 9) bot wit A, B, C, D may be confusing. why not use the a,b,c consistent with table 2?
- rielaborated? (caption Figure 9)

**References**

Rath, V.; Gonzalez-Rouco, J. F. Goosse, H., Impact of postglacial warming on borehole reconstructions of last millennium temperatures, Climate of the Past, 2012, 8, 1059-1066

Safanda, J. Rajver, D., Signature of the last ice age in the present subsurface temperatures in the Czech Republic and Slovenia, Global and Planetary Change, 2001, 29, 241-257

Bodri, L. Cermak, V., Climate changes of the last millennium inferred from borehole temperatures: results from the Czech Republic Part I, Global and Planetary Change, 1995, 98, 111-125

Bodri, L. Cermak, V., Borehole climatology: a new method how to reconstruct climate, Elsevier, 2007

Shen, P. Y.; Wang, K.; h. Beltrami Mareschal, J.-C., A comparative study of inverse methods for estimating climatic history from borehole temperature data Palaeogeogr., Palaeoclimatol. Palaeoecol. (GPC section), 1992, 98, 113-127
* * *

---

## Author Comment (AC1) · 23 Jul 2017

Dear Editor,

The replies to (in red italic) to all the reviewers comments are in the supplement file while in the attached files there are 4 new figures that we'll add to the manuscript and that are part of our replies. The numbering of these new figures is provisionally in roman numbers to distinguish by previous figures. In addition we are attaching also the new fig. 2 because is also part of the replies. All the other requested changes on

the original figures will be added directly in the revised text in the case that hopefully will be accepted for the final revision.

Please also note the supplement to this comment:
https://www.clim-past-discuss.net/cp-2017-23/cp-2017-23-AC1-supplement.pdf

[Figure]

**Fig. 1.** Fig. I Synthetic data for the present time. It is remarkable that also varying the alpha value by 33% the reconstructed GST does not vary significantly.

[Figure]

[Figure]

**Fig. 2.** Fig. II Synthetic data for three past years (2013,2014 and 2015).It can be seen that the inversion in the last 50 years is more accurate than the inversion of Fig I.

[Figure]

**Fig. 3.** Fig. III Example of different GSTh with different alpha with the extreme of heat flow values known for the region.

[Figure]

[Figure]

**Fig. 4.** Fig. IV Comparison of the temperature profiles of 2015 obtained in the case of a constant thermal diffusivity (facies a; red dots) andB) with a multilayers thermal diffusivities according the differen

[Figure]

[Figure]

**Fig. 5.** Figure 2. Topography of the SSB site: a) Digital Elevation Model (res. 5 m) of the SSB site and b) SSW-NNE and N-S transects through the Stelvio summit. Horizontal and vertical scales are the same.

**Supplement:**

**Reviewer 1**
**MAIN CONCERN: However, I do have some concerns regarding the procedure and method presented here – particular regarding uncertainty: the manuscript provides a too "straightforward" GST history reconstruction, neglecting any uncertainty ranges in the parameters involved. What about different regularization parameters? Could synthetic temperature profiles and the corresponding reconstruction of GST history give more insight into the method? Some additional (sensitivity) studies and a corresponding critical discussion is necessary, in my opinion. Data is available only at a few depth levels (Figure 5).**
**How about the minimum value at 60 m particular? What would be the influence on the result if possible variations of the data beneath and above this depths occur? Regarding the result in Figure 6, it is not possible to assess any uncertainties or to distinguish between effects arising from the "smoothed" data and the presented inversion.**
**As a consequence, I consider numerical methods for reconstructing GST history in this case (mountain area, unfrozen water content, uncertainties. . .) superior to analytical methods. If the latter ones are applied, a thorough justification and critical discussion must be given. Regarding this, presenting only one GST history result in Figure 6 and using this for the interpretation does not comply with the demands/conclusions of the manuscript.**

We have added the following discussion to the paper.

*3.4 Validation on synthetic data*

*In order to validate our GST inversion method we have generated a synthethic data set as follows. An ideal GST was chosen (dashed curve in Fig. I) and equation (1) was solved by a finite difference method with a spatial grid spacing of 1m. Homogeneous Neumann boundary conditions were imposed at the well bottom and the ideal GST as Dirichlet data at $z=0$, thus obtaining synthetic data for the measurements of temperature in the well. The computed temperatures were saved for the depth at which the real thermometers in SBB are located (see Sec 3.1), for the present time, as well as for 1,2 and 5 years before present. We then used the generated data as input to the inversion algorihtm described in the previous section and compared the reconstructed GST with the ideal one used to generate the synthetic data.*

*In the first experiment we fed our inversion algorithms only with the synthetic data for the present time. The value of alpha that best fits the exact GST is alpha=0.15, but in Figure I one can see that also varying this value by 33% the reconstructed GST does not vary significantly.*

*Next we fed the inversion algorithm also with the synthetic data for the past years. First, the inversion is expected to be more accurate since the algorithm can average not only on the temperature at a given depth but also on the variation of the temperature in the last years at that depth. Moreover, the algorithm should also be more robust, since it relies on a larger data set. Both these effects can be appreciated in Fig II, where it can be seen that the inversion in the last 50 years is more accurate than the inversion of Fig I and that a wider variation in the value of alpha is possible without affecting very much the quality of the reconstruction.*

Moreover we now present in Figure III the inversion of the real data obtained with different flux and alpha parameters. According to the knowledge of the heat flow in the area the values should vary between 85mWm$^{-2}$ and 70 mWm$^{-2}$ and therefore we selected three different values of alpha (0.8; 0.95 and 1.1) for each of the two values of heat flow.

Regarding the question raised on the minimum at 60 m of depth, it is true that as happened in a large part of the monitored permafrost boreholes there are fixed thermistors at several depths and not a repeated continuous logging and therefore the temperature profile is characterized also by the chosen depths.
In this case we can only say that the thermistor at 60 m did not have any instrumental problems during the monitoring period and it is located in the more widespread dolostone facies (facies a), so it is surely not influenced by layers of different thermal properties as shown also by the borehole stratigraphy of fig. 4.

**ABSTRACT:- Abstract: a significant part of the manuscript deals with the method used for GST reconstruction. Therefore, the method etc. should be mentioned in the abstract. Also, some important information one the borehole, such as depth, temperature ranges etc.**

*We agree with the reviewer and we'll add some sentences regarding the used methods and the main info on the borehole characteristics as requested.*

**Line 18: . . .roughly double the MAAT. . .is not clear, doubling of the increase of MAAT?**
*Thanks for the comment. The sentence will be correct as ... roughly doubling the increase of MAAT in the Alps.*

**- Line 22: linearly only, if there is additionally no heat production.**
*Thanks for the comment. The sentence now reads*

*If rock was homogeneous and no temperature change were to occur at the surface, the temperature would increase linearly with depth, unless spontaneous heat production is present on the vicinity of the well.*

**LINES 27-28: the propagation of signals is a diffusive process, therefore it is interesting where a maximum of a signal occurs.**
*It is true that it is interesting understand where the maximum of the signal occurs but the sentence here reported has been written just to give an idea of the velocity of the signal in the rock. Obviously it is very variable according all the variable of the heat diffusion law.*

**- Line 13: no groundwater flow only within continuous permafrost, more explanation needed.**
*May be the reviewer referred to line 31. We agree that more explanation needed and for this reason we change the sentence in:*
*Since normally no groundwater circulation is present within continuous permafrost in the polar areas but also in rocky areas within mountain permafrost, boreholes drilled in these areas are particularly suited for GST reconstructions.*

**- Line 41: the authors should justify this statement by some calculation (". . .for much of the last millenium"), due to the diffusive nature, the signal of the last millennium is not only visible in the upper 230 m, so a "truncation" of this signal may lead to a bias.**
*It is true so we change the sentence in "...it allows reconstruction of GST for some centuries and much more than in the other mountain permafrost boreholes.*

**Line 74: heat flow in mountainous areas may differ strongly from the typical literature values, so an estimation of permafrost thickness only using this value is questionable**
*It is true but in this chapter we are only citing what was calculated using literature values by previous authors that worked in the study area (Guglielmin, 2004). Moreover, considering the new results, permafrost thickness should be more than 240 m considering the thermal gradient in the lower part of the borehole currently measured and the temperature at 235 m of depth.*

**Line 87, laboratory data: there is no information about porosity, this parameter is important with respect to the latent heat effect. It is only mentioned that there is no evidence of ice encountered, although ice has been encountered in a very close borehole only a few meters apart. The temperature range (-2-0 ◦ C) is within the very range where a coexistence of both phases in soil/rock occurs (see references below).**
*It is true that porosity is important especially in frozen sediment, here the porosity of the dolostone is typically < 5% (Berra, personal communication) and therefore is practically negligible, only where karst phenomena occurred ice can be present.*

**Line 93: if the accuracy of the measurements of thermal properties is around 5%, it is then necessary to state values in table 2 accordingly (three decimal places are certainly**

**not applicable).**

*The same applies to Table 1 regarding the temperature gradients.*
*Yes it is true. We modified the values with one decimal.*

**Please check generally, if "◦ C" is used for absolute temperatures and "K" for temperature differences, this makes the distinction easier.** Done

**Line 126: the linear trend in the "deepest par" (below 60 m) can be still disturbed by a transient signal from the surface, so I does not really represent a background signal. This should be discussed.**

*We know that for borehole climatology a borehole of 235 m is considered a shallow borehole, nevertheless SSB borehole is the deeper borehole in mountain permafrost in the world and according GTN-P (Global Terrestrial Network-Permafrost within WMO) only 55 boreholes are deeper of our borehole on more than 1100 permafrost boreholes in the world. Clearly, the true steady state condition can not easily be estimated from the temperature and thermal properties alone for a shallow borehole, and therefore as often done in practice it has been assumed that the quasi-linear bottom part of the profile represents the steady state geothermal gradient. We know that as suggested for example by Rath et al., (2012) the warming after the LGM can affect this quasi-linear portion but we are not completely convinced that this correction can completely applicable to this case because GST for a large part of the Holocene (until probably 2000 years BP) Vedretta Piana glacier should overly the borehole.*

**Line 139: 0.2 is choses, why? What would be the effect different values? -**
The explanation is already present in the text at rows 140-141

**Line 147: it enhances the robustness: can this be justified?**

We believe that this can now be verified by the newly added sensitivity study. A reminder to section 3.4 is now added in the text.

**- Line 167: How is the optimal parameter alpha determined? What is the influence on the results for different values of alpha? The regularization has been applied in earlier works (see references below).**
As showed in fig. III the best fitting with the measured temperature and the heat flow estimated in the area is given by the alpha 0.95 and the heat flow of 0.07.
Numerical methods for the estimation of the regularization parameter, like GCV and L-curve, are not always reliable and in our case they provide an underestimation of the best value.

**- Line 208: what are the 13 depths listed in section 3.1? A figure would be helpful.**

*By mistake the depth at which the thermometers in SBB are located were not listed in the manuscript. We have added the relevant information to section 3.1.*

**- Figure 1 A and Figure 2: Scale is missing.**

*We added the scale at Fig. 1A while the scale of Figure 2A and 2B is already reported in Fig. 2B and is the same for both the panels as stated in the caption.*

**- Figure 3: labels are missing (time/temperature).**

*Done*

**- Figure 6: vertical axis is Delta T, referring to what?**

*It is already stated in the caption (reference period 1880-1960)*

**- Figure 7: What is 0.02? Is the linear regression really justified? How about correlation coefficients?**

*The referee is right and the label '0.02' is misleading. We changed the label with GST (ground surface temperature) because 0.02 was the depth of the most surficial thermistor (0.02 cm of depth) .The two linear regressions were deleted because were not statistical significant and because they were not explicitly cited in the text. Now in the text we added that the slight warming both of the air and of the GST was not statistically significant.*

**Suggested references: Mottaghy, D. & Rath, V. (2006), Latent heat effects in subsurface heat transport modelling and their impact on palaeotemperature reconstructions, Geophysical Jounal International 164, 236-245.**
**Romanovsky, V. E. and Osterkamp, T. E. (2000), Effects of unfrozen water on heat**
**and mass transport processes in the active layer and permafrost. Permafrost Periglac.**
**Process., 11: 219–239.**
**Rath, V. & Mottaghy, D. (2007), Smooth inversion for ground surface temperature histories: estimating the optimum regularization parameter by generalised cross-validation,Geophysical Journal International 171 (3), 1440-1448.**

*Thanks for pointing them out. The last one is very interesting and is now quoted in the paper. The other two are less relevant since the effect of ice/water content in rocks with such low porosities is negligible. Nevertheless, we included Mottaghy and Rath (2006) as example of the effects of latent heat on the paleotemperature reconstructions.*

**Reviewer 2 (V. Rath)**

**While the manuscript contains interesting data worth while publishing, I'm highly skeptical with regard to the particular approach taken by the authors. It leaves open several important questions, cannot be regarded as really novel, and is not well described. Therefore I think the article could only be published in CP after very thorough (major) revisions.**
*We hope that with the integrations and changes we solved the questions.*

**Major comments:**
**P2, L43-55:These paragraphs read as f the existence of permafrost is essential to the reconstruction of past surface temperatures from borehole temperature profiles, which is not correct. Furthermore the early work of Lachenbruch Marshall did not take into account the generic permafrost-related processes as freezing/thawing. This is related to the question of the existence of significant porosity (not even mentioned in the text). If water/ice-filled porosity is very small, the "dynamic" effects of permafrost are of course negligible. But then, the title may be a misnomer, and permafrost should be omitted there ("subsurface temperatures" instead of "permafrost temperatures").**
*We partially disagree with the reviewer because these paragraphs illustrate the state of the art regarding the use of permafrost boreholes profiles for GSTh. We don't think that in a paper like this one in which the porosity of the rock is very limited (<5%; see comments to reviewer 1) it is essential describe in the introduction also the factors that in general could be potentially affect the temperature profile but only which that are present here like topography.*
*Nevertheless we will add this sentence after row 41 "Several factors like porosity, water/ice and latent heat flows can influence significantly the thermal properties and the thermal signal measured in boreholes."*
*Moreover regarding the title we disagree with the referee's suggestion, because the thermal profile analysed here is recorded within permafrost, which is a particular thermal state of the material that is independent by the ice content. We thus don't think that "permafrost temperature" is misleading, but on the contrary it points out the relevance of this being the first ever thermal profile of some length analysed in the world mountain permafrost.*
**P2, L53-53: I disagree with the sentence regarding the importance of topographic effects. Even on a fully symmetric mountain these effects will be present. Moreover, the differences in insolation will produce an asymmetric regime. A N-S slice thus would**

**have been more relevant in Fig. 2, as this would be more characteristic with respect to the surface temperatures, and also would show more asymmetry. In order to be published, there should be a quantitiative assessment of the topographic effects with respect to the 1-D model used for inversion.**

*We agree with the reviewer 2 that even in a fully symmetrical mountain some effects can be present, depending on the slopes of the sides. However we point out that the topography around SBB is very flat and we thus believe that these effects would be below the instrumental sensitivity. We can show this with a "forward" simulation (i.e. computing and comparing borehole temperatures from an hypothetical GST in the 1D and 3D model), but we can do so only after the holidays. If it is considered mandatory, we will ask for a deadline extension.*

*Moreover we will change the figure 2 including also the suggested N-S transect that shows clearly that along this transect the slope is almost sub-horizontal.*

**P3, L83: Please comment on these literature values.**

*This paragraph referred to what was carried out in the study area so for us it is inappropriate to discuss in this section the literature values. In any case the heat flow values used in Guglielmin (2004) and reported in Cermak et al., (1992) are the maximum values valid for this area of the Alps indeed according Della Vedova et al., (1995) although there are no values exactly in the study area seems more reasonable the values of 70 mWm$^{-2}$. Regarding the thermal conductivity used in Guglielmin (2004) is very similar to what we found through the laboratory analyses of the more widespread facies of the dolostone that characterizes the study area. Moreover the calculated permafrost thickness by Guglielmin (2004) is not so far from the reality indeed he calculated 220 m and we found now that permafrost thickness is around 240 m.*

**P4, 3.2 Lab data: In table 2 there are three facies with quite different properties. How did you use this in your inversion? Note that a correct layered solution is given in Bodri & Cermak 1995.**

The differences of thermal properties are not so relevant respect the accuracy of the measurements except that for facies c that occurs in very thin and rare layers. Nevertheless we made now also a multilayered solution that we show in fig.IV and it is visible that the effects of these layers is negligible. The figure is a computation with the "forward model" in which we consider a (hypothetical but realistic) GST anomaly and compute the temperature anomaly by numerically discretizing equation (1). The GST employed is shown in the top-left panel, while the other panel shows the anomalies in the borehole temperature that would be caused by such GST for 2015. It is clear that the differences are well below the instrumental sensitivity.

**P4ff, 3.3 Theory: While the authors try do give a description of their approach, there are many claims or assumptions which need clarification. As the theory section is already rather long, it might be useful to put the details into an appendix, and concentrate on the essentials in the main text.**

*We agree and we will move part of the theory section in the appendix.*

**• Any assumptions on the physical limitations of the model should be mentioned in the text, e.g. porosity, latent heat release, properties regarded constant.**

*We agree and we explicitly written in the text that porosity is very limited (< 5%) and considering the temperature of the bedrock no latent heat flows are considered while thermal properties were considered variable according the laboratory values obtained for the main facies occurring in the bedrock.*

**Why and how detrend? Is this detrending unique? (L132ff). This trend surely does not only realut from regional heat flow density, bur includes all earlier events, which often leads to an approximately linear signal at these shallow depths (amongst others, Safanda & Rajver 2001, Rath et al. 2012). I would also not call it a detrending - it is a different nontrivial inverse problem for background heat flux. This is also related to the choice of the length of the temperature history to be reconstructed and the relevance of the resulting from the inversion.**

*As for the reply to the comment to L126 of the Reviewer 1 in practice often it has been assumed that the quasi-linear bottom part of the profile in shallow boreholes like SSB represents the steady state geothermal gradient.*

*To clarify we could change at L 133 that " coming from the heat flow and past climatic changes*

*since the Last Glacial Maximum as found for deeper boreholes by Safanda and Rajver (2001) or by Rath et al., (2012). Moreover the very good fitting of the inversion results for the interval of 500 years here chosen with the available other climatic reconstructions confirm the validity of our inversion*

• **Time lapse used in this inversion(L152-154)?**

*It is not clear what the reviewer asking. If he is referring to the time interval of the chosen points of the GSTh is well explained at L141-148. The time points are variable with a shorther span (i.e. less than 1 year between 2015 and 2005 to 8 years between 1515 and 1550 AD)*

• **This sentence is not comprehensible (L156-157).**

*We rephrased the sentence as" Not that the inverse problem for the heat equation (1) is well-known to be severely ill-posed, the matrix L is strongly ill-conditioned and its singular values decay exponentially to zero, with related singular vectors largely intersecting the subspace of high frequencies (Serra-Capizzano, 2004).*

• **Why is Tikhonov better than TSVD? A comparison figure would help. (L 164ff).**

*It is well-known that Tikhonov provides a better filtering with respect to TSVD [Hansen 1998] because it implements a smooth transition from low to high frequencies, while TSVD has a discrete parameter for cutting all the frequencies above a certain index.*
*Moreover, some preliminary tests have confirmed such statement.*

• **Which method was used to choose alpha for this study (L169ff)?**

*A detailed discussion on the choice of alpha and on the robustness of the algorithm with respect to this choice has been added to the paper in response to the first reviewer's comments.*

• **Smoothing regularizations have been used many times in the past (L175) - see Shen et al. 1992, Bodri & Cermak 2007, also the references given by Referee #1.**

*We change the sentence "The use of a regularization matrix R for this application is a novelty introduced in this paper" in "The use of a regularization matrix R for this application is a novelty although several other regularization smoothing parameters were already used (i.e. Shen et al., 1992; Rath, V. & Mottaghy, D. 2007 )*

• **This sentence is incomprehensible (L178-179).**

*If a vector x belongs to ker(R) then ||Rx||=0 and hence in the minimization problem (5) the penalization term disappear and the data are perfectly fitted.*

• **R is not square. Which solver is used? Boundary conditions in R? Is the condition of L180 fulfilled with your construction of L and choice of R? For R to be a discrete approximation to a differential operator a factor of (Dt)^2 is required (for constant Dt in L187.**

*The normal equations associated to (5) involve $R^TR$ . The solver is the GSVD (generalized singular value decomposition) implemented in the Matlab toolbox 'regtools' available at http://it.mathworks.com/matlabcentral/fileexchange/52-regtools*
*The condition L180 is always satisfied when L is an integral operator and R is a differential operator because ker(L) belongs to high frequencies while ker(R) belongs to low frequencies.*
*It is a scaled version of the discretization of the first derivative, the scaling factor is included in the regularization parameter.*

**General:**
**An assessment of uncertainty is missing, which is absolutely necessary particularly in the case of an ill posed problem. This is even more important when using a simple 1-D model which neglects so many effects which may bias the results.**
**While not improvingwith respect to the mentioned physical assumptions, already Monte Carlo studies and sensitivity considerations would help in this respect. Any result and interpretation**
**based on a single inversion can not be considered as reliable and is prone to bias.**
**The lack of a critical evaluation, which is essential when reconstructing ground surface temperature histories, makes it difficult to judge the value of the results obtained.**

*As written in the answers to the comments of reviewer 1 and by the new 4 figures and the addition to the figure 2b of the N-S profile we made some sensitivity and critical discussion of the used method and of the main physical parameters involved and therefore we hope that now the reviewer can be less skeptical and more convinced of the data treatment.*

**Minor items:**
• **Use dots, not commas for the decimal in the tables.**
DONE
• **Caption figure 3 (most important result of the study) should be informative.**
*Figure 3 is not meant to show any result but is merely an illustration of the functional form of the GST history of equation (2). Nevertheless the referee is right and the figure will be redone with axis labels (time for 'x' and GST anomaly for 'y') and its caption will be:*
*"Example of a GST history parametrized by equation (2)"*
• **Facies d in Figure 4 not in table 2.**
*Facies d was not analysed in laboratory because it is a local variation of the facies withouth any lateral continuity (from geological data of the area) and it is also the less common and present only in the first 42 m of depth.*
• **Marking facies (Figure 4) and paleoclimatic evens (Figure 9) bot wit A, B, C, D**
**may be confusing. why not use the a,b,c consistent with table 2?**
DONE
• **rielaborated? (caption Figure 9)**
DELETE
**References**
**Rath, V.; Gonzalez-Rouco, J. F. Goosse, H., Impact of postglacial warming on borehole reconstructions of last millennium temperatures, Climate of the Past, 2012, 8, 1059-1066**
**Safanda, J. Rajver, D., Signature of the last ice age in the present subsurface temperatures in the Czech Republic and Slovenia, Global and Planetary Change, 2001, 29, 241-257**
**Bodri, L. Cermak, V., Climate changes of the last millennium inferred from borehole temperatures: results from the Czech Republic Part I, Global and Planetary Change, 1995, 98, 111-125**
**Bodri, L. Cermak, V., Borehole climatology: a new method how to reconstruct climate, Elsevier, 2007.**
**Shen, P. Y.; Wang, K.; h. Beltrami Mareschal, J.-C., A comparative study of inverse methods for estimating climatic history from borehole temperature data Palaeogeogr., Palaeoclimatol. Palaeoecol. (GPC section), 1992, 98, 113-127**
*We will add the first three references while the fourth (that we know well, is a very good generic book on the borehole climatology) so it is difficult to include properly although we could include in the first part of the introduction.*

***New Figures Captions***

*Fig. I Synthetic data for the present time. It is remarkable that also varying the alpha value by 33% the reconstructed GST does not vary significantly.*
*Fig. II Synthetic data for three past years (2013,2014 and 2015).It can be seen that the inversion in the last 50 years is more accurate than the inversion of Fig I.*
*Fig. III Example of different GSTh with different alpha with the extreme of heat flow values known for the region.*
*Fig. IV Comparison of the temperature profiles of 2015 obtained in the case of a constant thermal diffusivity (facies a; red dots) andB) with a multilayers thermal diffusivities according the different facies and different temperature ranges in the borehole (blu dots) for the reconstructed GSTh.*

---

## Author Response (AR1)

[revised manuscript text omitted]
 two-dimensional model, in two vertical slabs corresponding to the two sections shown in Fig 2. The two-dimensional domains were meshed with the GMSH program and the heat equation was solved using linear Lagrange finite elements in space and backward Euler in time.

Figure 14 compares the temperature anomalies that each of the models would predict in the well at present time. The dots are the predicted well anomalies at the depth of the thermometers in SSB and the error bars correspond to their accuracy of 0.1°C. One can see that the predictions of the two-dimensional model with flat terrain coincide with those of the one-dimensional one. Furthermore, the two-dimensional model applied to the section with the steeper sides (the SSW-NNE one, blue line) gives rise to predictions that are within the instrumental error. Finally, the other section (the N-S one, red line), which has a flatter terrain, gives rise to predictions that are quite close to those of the one-dimensional model. A 3D model would of course give rise to predictions that would be in between the red and blue line in the figure and that would not be distinguished from the ones of the one-dimensional model by the thermometers in SSB.

Finally, let us remark that for the forward model, a numerical 2D simulation takes minutes to complete on a pc equipped with a Corei7 quad-core. Using a numerical multi-dimensional simulator in the inverse problem would of course require to compute several times the forward model and would thus take a lot longer than the few seconds in which our proposed method can compute the reconstructed GST depicted in Fig. 10.

Fig. 1

[Figure]

                                  Fig. 2

[Figure]

[Figure]

                               Fig.3

[Figure]

                               Fig.4

[Figure]

                               Fig. 5

[Figure]

Fig. 6

[Figure]

Fig. 7

[Figure]

                              Fig. 8

[Figure]

                              Fig. 9

[Figure]

Fig. 10

[Figure]

                                   Fig. 11

[Figure]

                                   Fig. 12

[Figure]

                                Fig. 13

[Figure]

                                Fig. 14

[Figure]

---

## Author Response (AR2)

Dear Editor,

Here there are the point by point replies of the comments of the reviewer 1 in red.

We hope that with this revision the paper can be considered acceptable for publication.

Prof. Mauro Guglielmin

The manuscript has been revised and I appreciate the answers/amendments the authors provide. However,
in my opinion, the concerns expressed the first previous review are only considered to a minimal extent,
still leaving some important questions.

The remaining points are:

- Line 83: the PACE borehole revealed some ice fills karst, but this is still not discussed further. Is it justified
to neglect this finding?

Yes, It is justified because the two small karst in reality were two fractures almost vertical crossed by the
first borehole for 1-2 meters filled by ice. More in general the examined area is avoided by the karst and
the fractures are generally not abundant. From some electrical resistivity soundings (not shown) carried out
in the area no traces of massive ice were found. To avoid possible confusion we change in the text and
instead of " small karst" we use the more appropriate term "fractures".

- In the figures, please state clearly what the different colors and line styles refer to.

DONE

- Figure 4,5 and Figure 8: the alpha value differs between the reconstructions using the synthetic and real
data, why? As seen in Figure 4 and 5, a variation can lead to significantly different results.

The regularization parameter alpha depends on the dataset and the noise level. Hence the three alpha
values used for the reconstructions with synthetic and real data are not related to each other.

- Fig. 9 is not cited in the manuscript text.

Sorry, there was an error because it was cited but with the wrong number (8 instead of 9). Now we correct
it but we expand more the sensitivity respect the thermal diffusivity as  follows " The effects of the
different thermal diffusivities measured in the different facies of the SSB borehole are also illustrated in
Figure 8 where is possible to notice that the difference of temperature *a posteriori* between a model with a
constant diffusivity equal to the average value of the facies a between 0°C and -1°C (red dots) and the
model with the different diffusivities for each different facies layer (blue dots) is absolutely negligible (<
0.02°C) in all the depths with the exception of the uppermost (20 m) where the difference is  higher but still
very low  (0.06°C).

- The caption of Fig. 9 is not clear – where do the temperature ranges come from? As seen from Fig. 14 they
probably represent the uncertainty in the temperature measurements.

Thanks for the comment. We rewrote the caption of Fig. 8 (ex 9) as follows: Fig. 8 Effects of different
thermal diffusivity used in the model. The temperature profiles *a posteriori* of 2015 obtained in the case of
a constant thermal diffusivity value of the more widespread facies (a) (red dots) and in the case of with a
multilayers thermal diffusivities according the different facies according Fig. 6 (blue dots). The bars indicate
the variations of the measured temperature in the same year.

Consequently, it would be interesting to investigate the effect of this temperature uncertainty on the
reconstructed GST history, not only showing it along with the temperature anomalies resulting from the
reconstructed GST history.

In order to investigate the effects of the accuracy of the measures and of their variations in the three
examined years on the reconstructed GST history we added the variations of the reconstructed GST history
using all the variation ranges of the measured temperature at all the depths in Fig. 10.  It is visible that the
pattern did not change and the different curves have a range of variations of a maximum of 0.3°C for the
period between 1500 and 1970 while is less accurate for the cold period of the 80[th].

- Please justify your answer to the comment to line 167: „Numerical methods for the estimation of the
regularization parameter, like GCV and L-curve, are not always reliable and in our case they provide an
underestimation of the best value."

Several mathematical papers in the literature (see e.g. [1-4] below) have proved that for every numerical
method for the estimation of the regularization parameter exists at least an example where it fails.

[1] Hansen, P.C., Jensen, T.K., Rodriguez, G.: An adaptive pruning algorithm for the discrete L-curve
criterion. J. Comput. Appl. Math. 198, 483–492 (2006)L. Reichel and G. Rodriguez.

[2] Old and new parameter choice rules for discrete ill-posed problems. Numer. Algorithms, 63(1):65-87,
2013.

[3] Hanke, M.: Limitations of the L-curve method in ill-posed problems. BIT 36, 287–301 (1996)

[4] Hansen, P.C.: Analysis of the discrete ill-posed problems by means of the L-curve. SIAM Rev. 34, 561–
580 (1992)

In our application, we observe that GCV and L-curve do not provide a reliable estimation of the
regularization parameter, in practice they provide an underestimation, because the reconstructed GST
history has too large and non-physical oscillations.

- Lines 250 to 253: regarding the uncertainty in the reconstruction (which still needs to be determined in
my opinion), is justified to give results in this precision (0.96 °C)?

Thanks for the comment. You are right. We modified the text as follows "Our reconstruction after the cold
GST anomaly, between 1906 and 1941 AD, shows a slightly positive peak (ca. 0.1°C) in 1930 and afterwards
a very unstable period with a first sharp decrease of temperature until 1989 (between -0.2 and -0.6°C) and
a second even sharper increase, reaching in 2011 the uppermost GST anomaly value of the last 500 years
(around 1°C).

Also the caption of the Fig. 10 is changed as follows: Figure 10. Comparison between the anomaly of the
mean annual GST reconstructed by SSB borehole (black thick line), its uncertainty range (red shaded) and
MAAT anomaly reconstructed for the European Alps by Christiansen and Ljungqvist (2011) (grey line with
dots; data available online at: ttps://www.ncdc.noaa.gov/paleo/study/12355) both respect the same
reference period (1880-1960).

- As mentioned the thermistor at the critical depth seemed to work properly, but this could have been
investigated by varying the data used in the inversion (sensitivity study, uncertainty ranges, see comment
above).

Thanks again, the uncertainty in the reconstruction due to the slight variations in the time (less than 0.1°C)
of the measured temperatures has been reported in the new Fig. 10 as a red shaded area and it is possible
to see that the uncertainty is similar to the accuracy of the thermistors between the 1500 and 1970 while is
less accurate for the cold period of the 80[th].

- 2D model: what would be the influence of the vertical extension of the model (background gradient…)

The model concerns temperature anomalies, that is differences between the observed temperatures and
the background gradient. This latter is assumed linear (as normally used in boreholes of this depth, i.e.
Bodri and Chermak, 2007) and is substracted from the measured data before applying the inversion. In the
case of the "forward model" simulations (figure 14), we compute the solution of the PDE (1) and its 2d and
3d extensions, which represent models for the anomaly and thus do not deal directly with the background
gradient. Moreover To satisfy the criticism on the 2D models as requested by the reviewer we calculate the
temperature distribution considering the 3D effect as reported below.

- I still consider a 3D model necessary in order to assess the 3D-effect on the temperature distribution,
although the authors provide arguments that this would not yield further information.

Although we believe that considering the morphology of the site this effect was not relevant we calculate
the temperature distribution considering the 3D-effect of the mountain using a DEM with a resolution of 10
m (the best available for the area), considering the same thermal properties and heat flow used for the
reconstruction of the GST and we report the results of this new calculation together the 2D already showed
in the previous version of the manuscript in the new figure 14.

It is possible to observe that as expected the 3D-effect at least in this case study is negligible.

Moreover we change also the text of appendix 2 as follows" …..

First we computed the solution of the one-dimensional model (1). Next we computed the solution of the
corresponding three-dimensional model in a computational domain of 400X400 m centered around SBB
and 500 m deep, whose the top surface was obtained from a DEM (with a resolution of 10m). Such domain
was discretized with the GMSH program and the heat equation was solved using linear Lagrange finite
elements in space and backward Euler in time. The mesh was refined until numerical convergence was
observed and in Figure 14 we present the results for a mesh with 1.3 million of tetrehedra. The numerical
simulations were performed with the HPC cluster of the Dipartimento di Matematica of the Università di
Torino.

Figure 14 compares the temperature anomalies that each of the models would predict at SSB at present
time. The red dots are the predicted well anomalies at the depth of the thermometers at SSB. One can see that the predictions of the two-dimensional model with flat terrain (blue line) almost coincide with those of the one-dimensional one. Furthermore, the two-dimensional model applied to the section with the steeper sides (the SSW-NNE one, orange line) gives rise to predictions that are within the instrumental error (±0.1°C) whereas the N-S section (red line), which has a flatter terrain, gives rise to predictions that are quite close to those of the one-dimensional model. The predictions of the 3D model (dashed black line) are very close to the 2D flat and the 2D N-S (with difference always < 0.03°C).

Finally, let us remark that for the forward model, a numerical 3D simulation takes hours to complete on 16 computing nodes of our HPC cluster. Using a numerical multi-dimensional simulator in the inverse problem would of course require to compute several times the forward model and would thus take a lot longer than the few seconds in which our proposed method can compute the reconstructed GST depicted in Fig. 10.

Minor:

- Please check again, if "°C" is used for absolute temperatures and "K" for temperature differences (and the gradient, Table 1), this makes the distinction easier. We used in all the text only °C degrees.

- Table 2: unit for thermal conductivity is not correct.

Thanks we correct it.

To summarize, the authors present interesting and valuable data which "deserves" a thorough investigation, which has not been done yet in my opinion. Therefore, I recommend another revision in order to do fully justice to the available data.

We hope that the reviewer can be satisfied by the sensitivity investigations regarding the effects of 3D, 2D models that clearly indicate that, at least, in this case the changes of temperature distribution with the morphological and topographical conditions are not relevant. This is possible here because respect other alpine mountain permafrost boreholes the location is less steep and asymmetrical and the summit area is much larger. Also the effects of the thermal diffusivity and of the alpha value were considered and in the first case is clear as the difference of thermal diffusivity related to the different facies of rocks and of their temperature are negligible too because the stratigraphy is very homogenous and the range of temperature is quite narrow and also because the porosity is very low.

[revised manuscript text omitted]

                                          Fig. 2

[Figure]

                               Fig.3

[Figure]

                               Fig.4

[Figure]

                                    Fig. 5

[Figure]

Fig. 6

[Figure]

Fig. 7

[Figure]

Fig. 8

[Figure]

Fig. 9

[Figure]

Fig. 10

[Figure]

                               Fig. 11

[Figure]

                               Fig. 12

[Figure]

                            Fig. 13

[Figure]

                            Fig. 14

[Figure]